# Tissue Characterization Using Synchrotron Radiation at 0.7 THz to 10.0 THz with Extended ATR Apparatus Techniques

**DOI:** 10.3390/s22218363

**Published:** 2022-10-31

**Authors:** Zoltan Vilagosh, Dominique Appadoo, Negin Foroughimehr, Reza Shams, David Sly, Saulius Juodkazis, Elena Ivanova, Andrew W. Wood

**Affiliations:** 1Australian Centre for Electromagnetic Bioeffects Research, Swinburne University of Technology, Melbourne 3122, Australia; 2THz/Far-IR Beamline, Australian Synchrotron, Melbourne 3168, Australia; 3School of Health Sciences, University of Technology Melbourne, Melbourne 3000, Australia; 4Optical Sciences Centre, Swinburne University of Technology, Melbourne 3122, Australia; 5STEM, School of Science, RMIT University, Melbourne 3000, Australia

**Keywords:** ATR, THz, water, synchrotron, biological tissues

## Abstract

The attenuated total reflection (ATR) apparatus, with an added partial reflection/partial transmission mode, was used to demonstrate a novel way of characterizing water-based substances at 0.7 to 10.0 THz at the Australian Synchrotron THz-far infrared beamline. The technique utilized a diamond-crystal-equipped ATR to track temperature-dependent changes in reflectance. A “crossover flare” feature in the spectral scan was noted, which appeared to be a characteristic of water and water-dominated compounds. A “quiet zone” feature was also seen, where no temperature-dependent variation in reflectance exists. The variation in these spectral features can be used as a signature for the presence of bound and bulk water. The method can also potentially identify the presence of fats and oils in a biological specimen. The technique requires minimal sample preparation and is non-destructive. The presented method has the promise to provide a novel, real-time, low-preparation, analytical method for investigating biological material, which offers avenues for rapid medical diagnosis and industrial analysis.

## 1. Introduction

The THz spectra of biological tissues and specimens, such as microscopic organisms, bacteria, and cultured cells, can yield information regarding tissue hydration, bound water content, the presence of melanin, and fat content [1,2,3]. Attenuated total reflection (ATR) is a well-established spectral technique. It has the advantage of minimal sample preparation and the capacity to access and manipulate the sample whilst the scan is being performed. Since ATR scanning is non-destructive, the sample can also be subsequently manipulated and re-examined.

The Australian Synchrotron (AS) THz beamline ATR apparatus has a diamond crystal with a fixed *θ* = 45° incoming beam angle. Although the AS beamline offers a spectral range of 0.5 THz to 20.0 THz, this study found that the practical range for water-dominated substances was 0.7 THz to 10 THz. The ATR configuration can be employed to investigate water-dominated biological samples at a wide range of THz frequencies, but the method requires a paradigm change [4]. Given that much of the published data on THz spectroscopy of biological materials are in the <1.5 THz range, the AS capabilities offer a significant expansion of the data available.

The refractive index (***n***) of water reduces from ~2.2 at 0.7 THz to ~1.5 at 10 THz [5], whilst the ***n*** of diamond remains stable at approximately 2.38 throughout the 0.710.0 THz range [6]. The ***n*** of water is ~1.68 at 5.0 THz, and, according to Snell’s law, this is the boundary between total reflection (***n*** < 1.68) and partial reflection/partial transmission (***n*** > 1.68) with a diamond crystal surface set at *θ* = 45°. The significance of this is that the reflection generated by the described AS ATR apparatus in the 0.7 THz to 10 THz range crosses at about 5.0 THz from “true ATR” reflection to a partial reflection/partial transmission (PRPT) mode at THz frequencies <5.0 THz [5]. An alternative way of understanding the transition is that, during total reflection in “true ATR” mode, an evanescent wave is formed. The penetration depth of the evanescent wave into the sample *Dp* is described by the equation [7]:(1)$$Dp=\frac{\lambda }{2\pi \left(n_{1}\right)\sqrt{(\sin^{2\;}\theta_{1\;\;}-(\frac{n_{2}}{n_{1}}{)}^{2})}}$$
where, ***n***_1_ and ***n***_2_ are the refractive index of the crystal and sample, respectively, ***λ*** is the wavelength of the incident radiation in free space, and *θ*_1_ is the angle of incidence of the radiation. Under conditions where the absorption within the sample is negligible, the energy of the evanescent wave is returned to the outgoing, reflected traveling wave. If, however, absorption is not negligible, the resultant ATR reflection is given by the more complicated Hansen equations [8], which consider ***n***, the absorption coefficient (**α**) of the sample, and also the polarization of the incident radiation.

As a practical observation, the ***n*** of water tends to change more slowly than **α**, both with frequency and temperature [5,9,10]. In addition, ***n*** does not have regions of rapid change such as the lines in an absorption spectrum. Thus, in “true ATR mode”, a change in reflection is overwhelmingly a marker of a change in **α**.

Equation (1) yields two important insights. Firstly, there is a limit for *Dp* where if:(2)$$\sin^{2\;}\theta_{1\;\;}\leq {\left(\frac{n_{2}}{n_{1}}\right)}^{2}$$
then Equation (1) does not yield a real number solution. The same analysis can also be arrived at using Snell’s law, which can be stated as:sin*θ*_1_ ***n***_1_/***n***_2_ = sin*θ*_2_(3)
where *θ*_1_ is the incident radiation angle in the first medium, ***n***_1_ and ***n***_2_ are the refractive indices of the first and second medium, and *θ*_2_ is the angle of the transmitted radiation in the second medium.

There is transmission of a traveling wave into the sample only if sin*θ*_1_ ***n***_1_/***n***_2_ < 1. If sin*θ*_1_ ***n***_1_/***n***_2_ > 1, the solution demands that sin*θ*_2_ > 1, and the equation breaks down. In the case where Snell’s equation delivers a real sin*θ*_2_, the proportion of the incident radiation that is reflected at the surface is described by the Fresnel equations.

The Fresnel equations have terms for the refractive index but not the absorption coefficient. Thus, in the circumstance where Snell’s equation holds, the reflection at the surface does not test **α**; of a sample, only ***n***_1_/***n***_2_. In the case of the AS ATR, with a diamond crystal of ***n*** = 2.38, and with an incident angle of 45° (sin*θ*_1_ = 0.707), the ratio of ***n***_1_/***n***_2_ needs to be <1.414 for sin*θ*_2_ to be ≤ 1. This becomes the boundary for “true ATR”. Since 2.38 × 0.707 = 1.68, samples with an ***n*** > 1.68 cannot be examined in “true ATR” mode with the described AS configuration [11]. The reflectance of samples with an ***n*** > 1.68 is subject to Snell’s law and becomes a partial reflection/partial transmission (PRPT) mode.

There is nothing exceptional about the 45° angle in the ATR configuration, but it does dictate the range at which the ratio ***n***_1_/***n***_2_ satisfies the requirements of Snell’s law. Since the refractive index of water varies significantly in the 0.7 THz to 10.0 THz range, if *θ*_1_ is changed, the frequency that forms the boundary between the “true ATR” and PRPT mode is also changed.

The angle of *θ*_1_ that forms the boundary between “true ATR” and PRPT in the range of 0.7 THz and 10.0 THz using water as the sample and a diamond crystal ATR is outlined in Table 1. The frequency-dependent changes for the refractive index of water rely on room temperature data from Segelstein [5].

The second insight from Equation (1) is that the *Dp* is directly proportional to ***λ***, the wavelength of incident radiation. At THz frequencies, the *Dp* is in the order of hundreds of microns for a diamond crystal ATR, with an incident angle of 45°. This becomes important in the sampling of inhomogeneous cellular structures which are found in biological tissues. Rather than sampling only several microns from the surface, as is the case for infrared-frequency ATR, the THz-frequency ATR apparatus samples many layers of cells. The ATR approach thus allows an averaged, homogeneous spectral sampling of the tissues. This opens the possibilities for investigating label-free, real-time cellular, tissue-level changes in living cells, multilayer and monolayer cultured cells, spores, and tissues taken from living organisms.

The THz-far infrared beamline at the Australian Synchrotron (AS) provides a high-brightness source, with the capacity to examine many samples in a short time. A THz spectral scan can be produced every 0.3 to 0.4 s. The sample stage on the ATR apparatus can be heated and cooled with an inbuilt system within 10 °C to 55 °C without modification. The advantage of a diamond crystal in the ATR apparatus for thermal studies is its superior thermal conductivity (five times higher than that of copper) and low thermal expansion coefficient [6]. This results in a very stable configuration.

The dielectric parameters of water at THz frequencies are highly temperature-dependent. The ***n*** and **α** of water change in opposite ways: ***n*** increases and **α** decreases as the temperature increases [9,10]. A decrease in **α** leads to less attenuation in reflection in the “true ATR” mode. Conversely, an increase in ***n*** makes the surface more reflective, increasing surface reflection in the partial reflection/partial transmission (PRPT) mode. Thus, with the AS ATR configuration, water, and samples dominated by water, have a pattern of a temperature-dependent increase in reflection as the temperature rises at frequencies under approximately <5 THz and a temperature-dependent reflection decrease at approximately >5 THz.

## 2. Materials and Methods

The reflectance of samples in the PRPT mode have been explored previously [4,11,12]. The two modes are illustrated in Figure 1.

Samples were placed on the ATR stage, directly onto the diamond crystal at the THz/Far-IR Beamline at the Australian Synchrotron, Melbourne, Victoria. The THz spectra were obtained using a Bruker IFS 125HR Fourier Transform Infrared spectrometer with an Si Bolometer (Bruker Optics, Ettlingen, Germany). The diamond prism had a quoted ***n*** of 2.38 [6] and an **α** of 0.1 to 0.2 cm^−1^ at 0.7 to 10 THz [12]. OPUS 8.0, Bruker Optics, build 8.0.19 (20180210) spectroscopy software for the measurement, processing, and evaluation of IR, NIR, and Raman spectra [13] was used for the initial data analysis. As noted previously, the great advantage of the ATR apparatus methodology is the simplicity of sample evaluation. The experimental layout is illustrated in Figure 2.

The sample was placed on the stage, and, if required, there was continual access to the sample for any manipulation that may have been needed. The temperature could be changed within the required thermal parameters using the integrated heating/cooling apparatus.

The samples used were pure water, water-based tissues (beef muscle, porcine muscle, fish muscle, porcine liver tissue), and water-based gel. Edible oils (olive and sesame), porcine adipose tissue, and petroleum gel were used to provide non-water-dominated samples for comparison. The fish muscle was from sp. Chrysophrys auratus (Australian snapper). The gel contained 90% water, VP/DMAPA acrylates copolymer, propylene glycol, triethanolamine, carbomer, and small quantities of other stabilizers. Petroleum gel is a mixture of hydrophobic mineral oils and waxes. The animal tissues were sourced from a commercial abattoir and food retail suppliers.

The sample was heated or cooled directly on the ATR sample stage. Although the sample on the ATR apparatus stage could be cooled with an inbuilt system to less than 10 °C, cooling to temperatures under 13 °C required more than 5 min, leading to the possibility of dehydration of the tissues on the sample stage. The resident heating system could raise the temperature at the ATR stage to above 55 °C within 5 min; however, there was concern that temperatures above 44 °C would lead to excessive tissue denaturation and release of water from the tissues.

Temperature stabilization at exactly 13 °C and 44 °C proved impractical, and the policy of first cooling the sample in place, followed by starting the scans at 14 °C (± 1°) and scanning continuously as the temperature rose, and then utilizing the scan taken at 44 °C (± 1°), was adopted. The scans in the interval between 14 °C (± 1°) and 44 °C (± 1°) are not reported in this study.

Each individual scan took approximately 0.4 s. The full process of heating and scanning from 14 °C to 44 °C took approximately 3 min per sample. The data were collected using 20 individual scans as a group starting at 14 °C (± 1°) and ending at 44 °C (± 1°). The 20 individual scans took approximately 8 s to complete. The data from the 20 individual scans at the set temperature were averaged within the OPUS software into a single spectral scan dataset at the two temperature points.

Control scans employing water as the sample using 50 and 100 individual scans per group showed no significant difference between these and the 20 individual scans per group, and since the temperature was easier to control for shorter time intervals, the 20 individual scans per group was adopted as the standard.

The reflection data were collected simultaneously at 0.0289 THz intervals in the 0.7 THz to 10 THz range, giving a total of 320 data points for every scan group. The raw spectra were normalized to air reflection = 100% for analysis.

## 3. Results

The effect of temperature variation within the apparatus was investigated by having a set of 14 °C and 44 °C spectra with an open stage (i.e., with only air at the diamond crystal interface). The results are presented in Figure 3. There was a significant difference in the air reflection output between 14 °C and 44 °C, which necessitated the normalization of tissue spectral data to the air spectrum at the same temperature as the sample, i.e., spectra at 14 °C were normalized to the 14 °C air spectrum and spectra at 44 °C were normalized to the air spectrum acquired at 44 °C.

Based on the temperature variation within the apparatus data, the <0.7 THz region was omitted from the analysis, as contrast in the <0.7 THz region was very poor, and the normalized reflectance proved to be erratic. The 0.7 THz to 2.0 THz data remains low contrast but was included since much of the published data on THz interaction with biological samples are in the <1.5 THz range. The >10.0 THz data were also disregarded due to the instrument-caused anomaly in the spectral data in the region of 11 THz. The measurement error, estimated by repeat spectral scans at the same temperature, was ±5% at the 0.7 THz to 1.0 THz region, reducing to ±0.2% in the 3.0 THz to 6.0 THz region and increasing to ±1.5% in the 9.0 THz to 10.0 THz region.

The data for pure water, water-based tissues, and water-based gel are presented in Figure 4. The reflectance data for all samples were normalized to air = 100%. All the water-dominated samples showed a similar pattern, with the temperature-dependent reflection changes (in PRPT mode) following the anticipated main dependence on ***n*** at <5.0 THz, and the temperature-dependent reflection changes above 5.0 THz being chiefly dependent on **α**, in the region of “true ATR”. The crossover “quiet point” proved to be at 5.2 THz. There was a region either side of the “quiet point” where there was a gradation of ***n*** vs. **α** influence. Fish muscle had a higher reflectance when compared to other tissues in the < 4 THz range. The crossover “quiet point” was, however, in the same region as the other tissues.

In the region around 5.2 THz, a “quiet point” is noted, where there is constriction in the temperature-change-generated dispersion in the spectral lines. This is the region where the relative influences between ***n***-dominated change and **α**-dominated change cross over in the vicinity of the boundary between the PRPT and “true ATR” modes.

The reflectance of the “quiet point” varied with the sample, with water having the lowest reflectance. The spectral scans of tissue and the gel have near-identical form to water, but they always have a higher reflectance for a given temperature, i.e., they behave as if they were at a lower temperature than would be expected if bulk water reflectance was the only constraint.

The “crossover flare” feature is best described as the spectroscopic representation of the more impressive part in the region where the spectral change is dominated by the absorption coefficient. After a region of relatively little change in the refractive index region, a widening (“crossover flare”) of the temperature-dependent reflectance range is noted.

Reflectance data for porcine adipose tissue, olive oil, sesame oil, and petroleum gel are presented in Figure 5. These data were also normalized to air = 100%. The olive oil, sesame oil, and petroleum gel samples do not have the water-like “crossover flare” or “quiet point” feature. From the literature [14,15,16], the refractive index of these substances is in the 1.45 to 1.55 range at THz frequencies. This results in olive oil, sesame oil, and petroleum gel being examined in “true ATR” mode throughout the range. Olive oil and sesame oil exhibit temperature-dependent reflectance increase at <2.0 THz, and, since the ATR apparatus is operating in “true ATR” mode, this indicates a temperature-dependent **α** increase in that region. This matches observations by other investigators [17]. The temperature-dependent spectra of olive oil and sesame oil show a wide divergence between the 14 °C and 44 °C in the 1.0 THz to 2.0 THz range, and an uneven reflectance gradient throughout the 0.7 THz to 10.0 THz range.

Porcine adipose tissue exhibits features of both “water type” and “oil type” spectra. There is a temperature-dependent “crossover flare” and “quiet point” feature, with the “quiet point” being reached at 3.5 THz, indicating that this is the approximate point where ***n*** = 1.68 for the sample. At frequencies under 2.0 THz, the porcine adipose tissue temperature-dependent spectral signature resembles that of the edible oils, albeit at a lower reflectivity, with variations in reflectivity mirroring the oil spectra.

## 4. Discussion

All tissues exhibited a general tendency to higher reflectance in the “true ATR” region (>5 THz) and a lower reflectance in the partial reflection/partial transmission region (<5 THz). The reflectance of oils (which, by virtue of their refractive index, were sampled in “true ATR” mode throughout the 0.7 THz to 10.0 THz range) showed similar variations in the <2 THz range, with peaks at 1.3, 2.0, and 2.5 THz. Oils exhibited a temperature variation in the <2 THz range which did not persist at higher frequencies.

When using water-based samples, there is no sudden change in the reflectance of any sample as it crosses over from partial reflection/partial transmission to “true ATR”. In fact, it is impossible to tell where this point may be from a scan at a single temperature. The temperature variation in the spectral scans shows only a slow transition from ***n***-dominant to **α**-dominant frequencies. Unequivocal **α** dominance was not achieved until 7 to 8 THz, which corresponds to a water ***n*** of approximately 1.5. This suggests that “true ATR” at THz frequencies is only unequivocally a marker of **α** when the sin^2^*θ* term in Equation (1) is 1.25 times larger than the (***n***_1_*/**n***_2_)^2^ term in the equation.

All water-dominant tissues and the water-based gel showed spectral features closely resembling pure water. The only exception was fish muscle which showed features similar to oil spectra at <2.5 THz and a higher reflectance in the <4.0 THz range. Fish muscle spectra were possibly influenced by their oil content. The spectral features of water-dominant tissues and the water-based gel resembled that of pure water, but with a consistently higher reflectance. This is consistent with the presence of bound water. Due to the reduced freedom of movement, the spectrum of bound water behaves as if it were a lower temperature than bulk water [3,18].

The crossover between ***n*** dominance to **α** dominance is marked by a temperature invariant “quiet point”. This region can be used as a reference point when comparing spectra from specimens which have a different temperature, or specimens where the temperature is not known.

## 5. Conclusions

An ATR apparatus-based method is presented which utilizes the anomaly of the large change in the refractive index of water at THz frequencies to extend the capabilities of a diamond-crystal-equipped ATR to a partial reflection/partial transmission (PRPT) mode. The derived spectra taken at temperatures of 14 °C and 44 °C, over the range of 0.5 THz to 10.0 THz, yield information regarding the bulk water and bound water content of biological substances. The combination of these elements is shown to be able to characterize the presence of water-dominant features including bound water and features indicating the presence of lipids. With the juxtaposition of the non-water-dominant spectral features, the possibility of a tissue-characteristic, temperature-associated spectral pattern emerges. The technique allows for the spectral pattern to be derived with minimal sample preparation and access to the sample at all times.

The “quiet point” feature can be used to compare disparate spectral scans derived at different temperatures at the frequency where the “quiet point” occurs. The frequency at which this point is found can be changed by altering *θ***_1_**, the angle of the incident beam, since the point is dependent on sin*θ***_1_**. This allows for a tunable point for comparison of spectral scans derived at different temperatures. As a practical measure, from Table 1 it is apparent that there is little change in the refractive index of water between 6.0 THz and 10.0 THz, making the technique less useful in that range.

The presented method requires further evaluation but has the promise to provide a novel, real-time, low-preparation, analytical method for investigating biological material, which offers avenues for rapid medical diagnosis and industrial analysis.

## Figures and Tables

**Figure 1 sensors-22-08363-f001:**
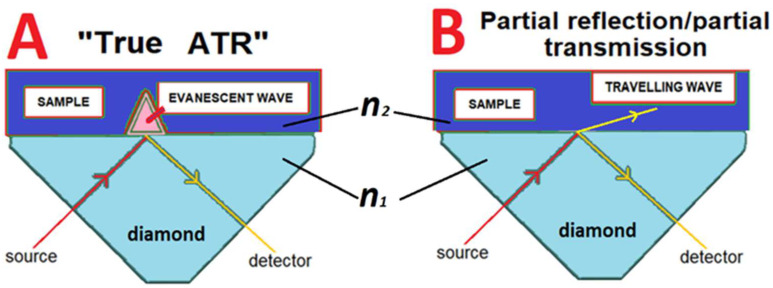
ATR apparatus configuration. (**A**) “True ATR” mode, with total reflection at the diamond/sample interface, producing an evanescent wave; (**B**) partial reflection/partial transmission mode, where some of the signal is transmitted into the sample at the diamond (***n***_1_)/sample (***n***_2_) interface as a continuing travelling wave.

**Figure 2 sensors-22-08363-f002:**
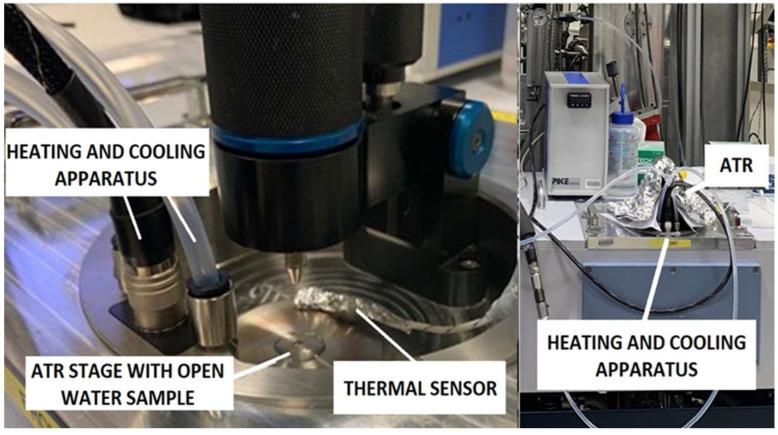
Experimental layout. The foil cover shown in the illustration on the right was used to aid in the initial cooling of the apparatus to 14 °C and to aid sample temperature uniformity during the heating phase.

**Figure 3 sensors-22-08363-f003:**
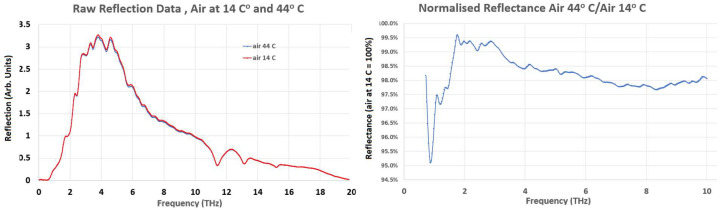
The effect of temperature variation on ATR spectra at 14 °C and 44 °C with no specimen in place. The variation between the 14 °C and 44 °C spectra was not uniform, varying by 0.5% to 5%, depending on the frequency.

**Figure 4 sensors-22-08363-f004:**
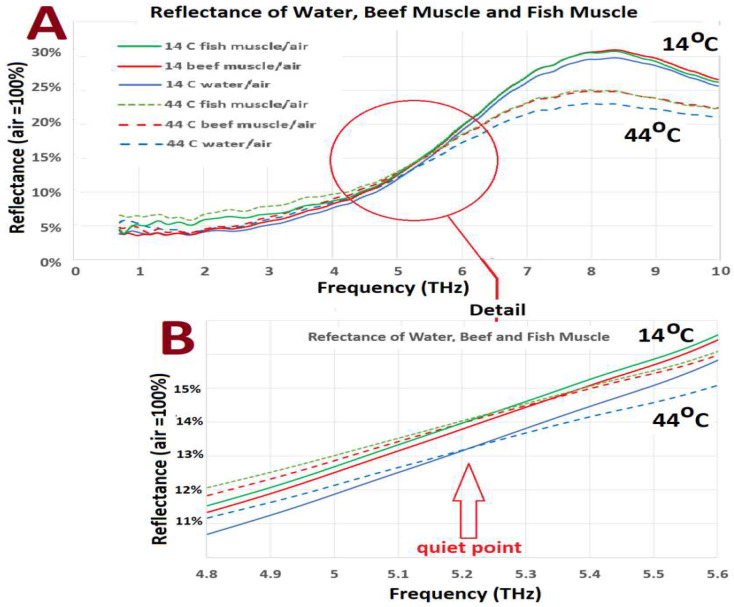
(**A**) The reflectance spectral data for water, beef, and fish muscle. (**B**) Detail of the thermal crossover “quiet point” for water, beef, and fish muscle. (**C**) Spectral data for water and porcine tissues. (**D**) The reflectance data for water-based gel. The frequency for the crossover “quiet point” is similar in all cases. In all cases, the pure water reflectance is lower than that for the tissues. The fish data show a significant divergence from the other tissues in the partial reflection/partial transmission part of the dataset. A widening (“crossover flare”) of the temperature-dependent reflectance range is noted as the spectrum crosses from the refractive-index-dominated zone to the absorption-dominated region.

**Figure 5 sensors-22-08363-f005:**
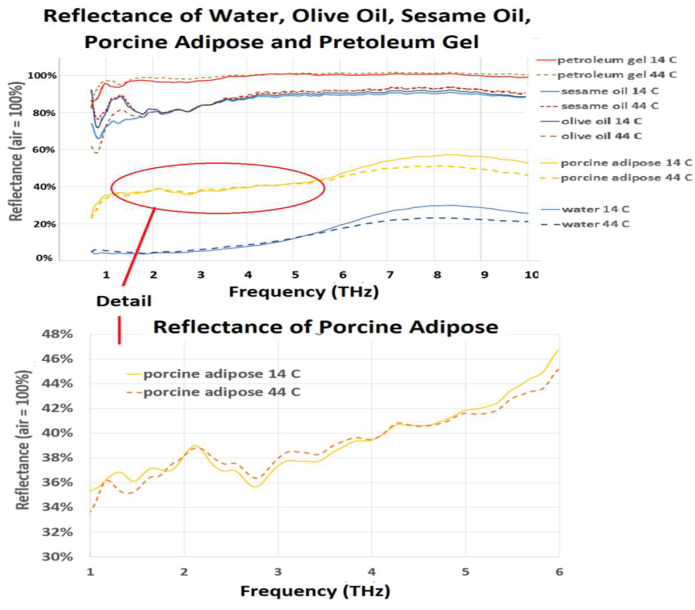
Reflectance data for porcine adipose tissue, olive oil, sesame oil, and petroleum gel (**top**), with detail in the 1.0 THz to 4.5 THz range shown on the (**bottom**) for porcine adipose tissue. The spectral data for the oil specimens show a high reflectance, in keeping with their low **α**. There is temperature variation in reflectance noted at frequencies < 2.0 for these samples. The reflectance data for petroleum gel are stable and featureless. Porcine adipose tissue shows features of both water content and fat content in its spectrum.

**Table 1 sensors-22-08363-t001:** Calculated angle for *θ*_1_ of boundary between “true” ATR and PTPR modes of water with a diamond crystal (***n***_1_ = 2.38).

Frequency	Refractive Index of Water *	*θ* _1_
THz	*n* _2_	Degrees
0.7	2.15	65
1.0	2.05	59
1.5	1.97	56
2.0	1.93	54
2.5	1.91	53
3.0	1.89	53
4.0	1.80	49
5.0	1.67	45
6.0	1.54	40
8.0	1.48	38
10.0	1.53	40

* Data from Segelstein [5].

## Data Availability

Not applicable.

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
