# Peer review of "Tissue Characterization Using Synchrotron Radiation at 0.7 THz to 10.0 THz with Extended ATR Apparatus Techniques"

_sensors, 2022, doi:10.3390/s22218363_

Round 1

Reviewer 1 Report

The manuscript presents the attenuated total reflection (ATR) technique with an added partial reflection/partial transmission mode as a novel way of characterizing water based substances. Authors demonstrate the data on registered spectra for various samples for several temperatures of 14o C and 44o C in the range of 0.5 THz to 10.0 THz/ Authors suppose this approach would enable detection of the presence of bound and bulk water in the samples. The benefits are the easiness of utilization and minimal sample preparation.

The article is written in a clear manner and is easy to understand. Introduction and Materials and Methods sections are detailed enough to help the reader to immerse oneself in the subject.

However I have minor comments.

1.      The reference for equation (1) should be given.

2.      Double “sin” word is met in lines 75-76 – “if sin 75 sin θ1 n1 / n2 < 1”.

3.      Figure 4. Curves are hard to distinguish. I would suggest adding numerical markers pointing each curve (e.g. curve 1, curve 2). Markers like triangles and circles are too small and cannot be distinguished. Color identification is insufficient also for A, B, and C subplots.

4.      Figure 5. The same comment. Figure quality of raster image is poor. Circle and cross markers are undistinguishable. Vector format for images should be considered.

5.      An error bar is a good practice for indicating on graphs with experimental data. The accuracy can be either given in the text.

6.      Can the comparison of the obtained results be made with similar studies using ATR technique?

7.      Authors state that the frequency at which this point is found can be changed by altering θ1, the angle of the incident beam, since the point is dependent on sin θ1. What range of this frequency change can be achieved. What are the geometry general restrictions and that of the set-up reported in the manuscript?

Author Response

Dear Editors and Reviewers

Sensors, MDPI

Please find attached the authors’ response to the reviewers’ helpful suggestions and criticisms the article titled:

Tissue characterization using synchrotron radiation at 0.7 THz to 10.0 THz with extended ATR apparatus techniques

We have made changes in the article in response. I hope you now find the article worthy of publication,

Yours sincerely,

Zoltan Vilagosh

Reply to comments and suggestions

Reviewer 1:

  1. The reference for equation (1) should be given.Re

ference added : Ahmad, M.; and Hench, L.L. Effect of taper geometries and launch angle on evanescent wave penetration depth in optical fibers. Biosensors and Bioelectronics, 2005. Volume 20(7), pp.1312-1319.

  1. Double “sin” word is met in lines 75-76 – “if sin 75 sin θ1 n1 / n2 < 1”.

corrected

  1. Figure 4. Curves are hard to distinguish. I would suggest adding numerical markers pointing → each curve (e.g. curve 1, curve 2). Markers like triangles and circles are too small and cannot be distinguished. Color identification is insufficient also for A, B, and C subplots.

Figures 4  and 5 have been revised and enlarged to improve readability

  1. Figure 5. The same comment. Figure quality of raster image is poor. Circle and cross markers are undistinguishable. Vector format for images should be considered.

Figures 4  and 5 have been revised and enlarged to improve readability

  1. An error bar is a good practice for indicating on graphs with experimental data. The accuracy can be either given in the text.

Accuracy given in text

  1. Can the comparison of the obtained results be made with similar studies using ATR technique?

To the authors’ knowledge, no similar results are available regarding features such as the “quiet point”. The temperature dependent absorption coefficient water in the 5 THz  to 10 THz region (where absorption coefficient dominates reflectance) is compatible with those cited by other authors (e.g. Zelsmann, H.R., 1995. Temperature dependence of the optical constants for liquid H2O and D2O in the far IR region. Journal of molecular structure, 350(2), pp.95-114. We have added a note in the text and a reference that effect.

  1. Authors state that the frequency at which this point is found can be changed by altering θ1, the angle of the incident beam, since the point is dependent on sin θ1. What range of this frequency change can be achieved. What are the geometry general restrictions and that of the set-up

We have added a note and a table to explain the geometry restrictions.

Reviewer 2 Report

This manuscript is devoted to the demonstration of a novel approach to characterizing water based substances at 0.7 to 10.0 THz with using the Australian Synchrotron with an added partial reflection/partial transmission mode. This approach shows the variation of the spectral features (a “crossover flare” feature appeared as characteristic of water and water dominated compounds and a “quiet zone” feature, where no temperature dependent variation in reflectance exists) that can be used as a criteria for the presence of bound and bulk water. These results can be interesting for scientific groups in areas of, spectroscopy, biology, medicine etc.

There are some points to correct or to make the information more clear:

1)      Authors investigated the water based samples at two various temperatures. But there are no analytical temperature dependences for refractive indexes and absorption of water and air in theoretical parts of manuscript. 

2)      It is necessary to write all formulas completely. E.g. Dp=equation (1)

3)      There are some errata in the text (e.g.”… if sin sin θ1 n1 / n2 < 1.” (75th – 76th lines); “…“quiet point” point feature” (232nd line) – it must be possibly “…“crossover flare” and “quiet point” features…”).

4)      The abbreviations of “Attenuated total reflection (ATR)” (19th and 36th lines) and “partial reflection/partial transmission (PRPT)” entered twice (86th and 267th lines)

5)      It is necessary to describe that the OPUS 8.0 software  (126th line) is the Bruker software for state-of-the-art measurement, processing and evaluation of FTIR, NIR and Raman Spectra with reference.

6)      There is the note concerned the Figure 4. The captions of axes and the legends especially are small. Besides these Figure’s parts have very low resolution.  All markers presented in Figure 4B look similarly. The curves in other parts of Figure 4 look like ones without markers. It is possible to write the color lines in legends without markers.

7)      There is the note concerned the Figure 5. The legends in the left part of Figure 5 require zooming for reading. The part of Figure presented in right side (named “Detail) must be marked in the left part of Figure 5.  Besides, authors write: “There is a temperature dependent “crossover flare” and “quiet point” point feature, with the “quiet point” being reached at 3.5 THz…”(232nd -233rd lines). But if the “quiet point” is discussed in text and described upper (in Figure 4B), the term of “crossover flare” applied to these experimental data is not discussed.

Except some criticisms for which some corrections have to be made, the text is satisfactorily written and describes with details the approach to characterizing water based substances at 0.7 to 10.0 THz with an added partial reflection/partial transmission mode

The manuscript can be published after minor revisions.

Author Response

Dear Editors and Reviewers

Sensors, MDPI

Please find attached the authors’ response to the reviewers’ helpful suggestions and criticisms the article titled:

Tissue characterization using synchrotron radiation at 0.7 THz to 10.0 THz with extended ATR apparatus techniques

We have made changes in the article in response. I hope you now find the article worthy of publication,

Yours sincerely,

Zoltan Vilagosh

Reply to comments and suggestions

 Reviewer 2:

  1. Authors investigated the water based samples at two various temperatures. But there are no analytical temperature dependences for refractive indexes and absorption of water and air in theoretical parts of manuscript.

All the temperature dependencies used are from direct experiment and from reference to literature. The paper did not seek to apply an analytical model.

  1. It is necessary to write all formulas completely. E.g. Dp=equation (1)

The “Dp” was left out in error, Corrected

  1. There are some errata in the text (e.g.”… if sin sin θ1 n1 / n2 < 1.” (75th – 76th lines); “…“quiet point” point feature” (232nd line) – it must be possibly “…“crossover flare” and “quiet point” features…”).

Corrected

  1. The abbreviations of “Attenuated total reflection (ATR)” (19th and 36th lines) and “partial reflection/partial transmission (PRPT)” entered twice (86th and 267th lines)

The authors felt that, due to the novelty of the term PRPT, and the distance between the two instances, the abbreviations should be explained a second time to aid the reader.

  1. It is necessary to describe that the OPUS 8.0 software (126th line) is the Bruker software for state-of-the-art measurement, processing and evaluation of FTIR, NIR and Raman Spectra with reference.

Added

  1. There is the note concerned the Figure 4. The captions of axes and the legends especially are small. Besides these Figure’s parts have very low resolution. All markers presented in Figure 4B look similarly. The curves in other parts of Figure 4 look like ones without markers. It is possible to write the color lines in legends without markers.

Figures 4 and 5 have been enlarged and improved to aid readability

  1. There is the note concerned the Figure 5. The legends in the left part of Figure 5 require zooming for reading. The part of Figure presented in right side (named “Detail) must be marked in the left part of Figure 5. Besides, authors write: “There is a temperature dependent “crossover flare” and “quiet point” point feature, with the “quiet point” being reached at 3.5 THz…”(232nd -233rd lines). But if the “quiet point” is discussed in text and described upper (in Figure 4B), the term of “crossover flare” applied to these experimental data is not discussed.

A note explaining the “crossover flare” has been added, it is essentially a phenomenon where reflectance undergoes a grater change with changes in the absorption coefficient  compared to the change in the refractive index.

Reviewer 3 Report

The authors proposed a method to distinguish the bound water and bulk water in biological tissues by an attenuated total reflection (ATR) apparatus. The technique utilizes a diamond crystal equipped ATR to track temperature dependent changes in reflectance. The manuscript is clearly stated, the data are detailed, and the experimental results and analysis are correct. It is suggested to publish after minor revision.

Suggestions and questions

1.    The figures and the text in the pictures are too small. Please enlarge the figures and increase the font size.

2.    The penetration depth of evanescent wave into the sample is related to the sample’s refractive index. How to use the depth of evanescent wave to explain the "crossover flare" feature in the spectra?

3.    Since the sample needs to be heated during the test, how to use the method proposed by the author to achieve rapid rapid medical diagnosis and industrial analysis?

Author Response

Dear Editors and Reviewers

Sensors, MDPI

Please find attached the authors’ response to the reviewers’ helpful suggestions and criticisms the article titled:

Tissue characterization using synchrotron radiation at 0.7 THz to 10.0 THz with extended ATR apparatus techniques

We have made changes in the article in response. I hope you now find the article worthy of publication,

Yours sincerely,

Zoltan Vilagosh

Reply to Reviewer 3

Suggestions and questions

  1. The figures and the text in the pictures are too small. Please enlarge the figures and increase the font size.

The figures 4 and 5 have been enlarged and improved to air readability.

  1. The penetration depth of evanescent wave into the sample is related to the sample’s refractive index. How to use the depth of evanescent wave to explain the "crossover flare" feature in the spectra?

The "crossover flare" feature is due to the greater change in reflectance with temperature when the ATR apparatus is performing in a “true ATR” mode, i.e. when it is (mainly) sampling the absorption coefficient rather the reflective index. The evanescent wave penetration depth equation does, indeed have a term for the refractive index, and the temperature dependent variation of the refractive index of water would be a factor in the change of the penetration depth, however the change due to the absorption coefficient is much greater and this is why it is the dominant factor in the "crossover flare".  We have added a not in the manuscript explaining the "crossover flare" feature.

The paper uses the penetration depth equation to illustrate the point that both this and Snell’s equations lead to the same conclusion – that ATR apparatus can be used in both “true ATR” and partial reflection, partial transmission mode. It is only the serendipity of the refractive index of a diamond crystal, and the changing refractive index of water in the 0.7 THz to 10 THz range that gives us this possibility.  

  1. Since the sample needs to be heated during the test, how to use the method proposed by the author to achieve rapid rapid medical diagnosis and industrial analysis?

All one needs to do to a water dominated sample is to put in on diamond crystal ATR and heat or cool by a quantifiable amount that is sufficient to produce contrast. There is no need for any other preparation. To that end, it then becomes a matter of ? faster than something else. The method is certainly faster than, many other tissue interrogation methods such as say histological fixation or even frozen section. It is an application that may find a niche.